# Chain of Reasoning for Visual Question Answering

**Chenfei Wu,**[∗] **Jinlai Liu**[∗]**, Xiaojie Wang, Xuan Dong**
Center for Intelligence Science and Technology
Beijing University of Posts and Telecommunications
{wuchenfei,liujinlai, xjwang, dongxuan8811}@bupt.edu.cn

## Abstract

Reasoning plays an essential role in *Visual Question Answering* (VQA). Multi-step and dynamic reasoning is often necessary for answering complex questions. For example, a question "*What is placed next to the bus on the right of the picture?*" talks about a compound object "*bus on the right,*" which is generated by the relation *<bus, on the right of, picture>*. Furthermore, a new relation including this compound object *<sign, next to, bus on the right>* is then required to infer the answer. However, previous methods support either one-step or static reasoning, without updating relations or generating compound objects. This paper proposes a novel reasoning model for addressing these problems. A *chain of reasoning* (CoR) is constructed for supporting multi-step and dynamic reasoning on changed relations and objects. In detail, iteratively, the relational reasoning operations form new relations between objects, and the object refining operations generate new compound objects from relations. We achieve new state-of-the-art results on four publicly available datasets. The visualization of the chain of reasoning illustrates the progress that the CoR generates new compound objects that lead to the answer of the question step by step.

## 1 Introduction

"The technical issues of acquiring knowledge, representing it, and using it appropriately to construct and explain lines-of-reasoning, are important problems in the design of knowledge-based systems, which illuminates the art of Artificial Intelligence" [1]. Advances in image and language processing have developed powerful tools on knowledge representation, such as long short-term memory (LSTM) [2] and convolutional neural network (CNN) [3]. However, it is still a challenge to construct "lines-of-reasoning" with these representations for different tasks. This paper meets the challenge in visual question answering, a typical field of Artificial Intelligence.

*Visual question answering* (VQA) aims to select an answer given an image and a related question. The left part of Fig. 1 gives an example of the image and the question. Lots of work has been done on this task in recent years. Among them, reasoning, named in different ways, plays a critical role. Most of existing VQA models that enable reasoning can be divided into three categories. Firstly, relation-based method [4] views reasoning procedure as relational reasoning. It calculates the relations between image regions to infer the answer in one-step. However, one-step relational reasoning can only construct pairwise relations between initial objects, which is not always sufficient for complex questions. It is not a trivial problem to extend one-step reasoning to multi-step because of the exponential increase of computational complexity. Secondly, attention-based methods [5, 6] view reasoning procedure as to update the attention distribution on objects, such as image regions or bounding boxes, so as to gradually infer the answer. However, no matter how many times the attention distributions are updated, the objects are still from the original input, and the entire reasoning

---

[∗]The first two authors contributed equally.

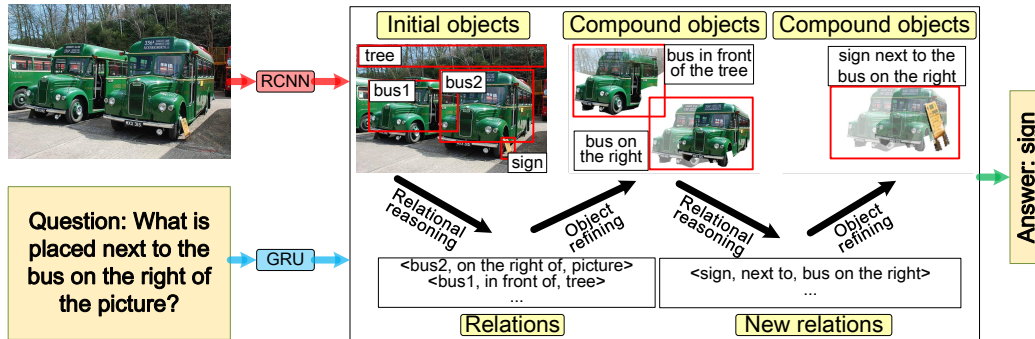

Figure 1: Chain of Reasoning for VQA. The alternate updating of objects and relations forms a chain of reasoning. The relational reasoning operation forms new relations between objects. The object refining operation generates new compound objects from relations.

procedure does not produce compound objects, such as *"sign next to the bus on the right"*, which many questions talk about. Thirdly, module-based methods [7, 8, 9] view reasoning procedure as a layout generated from manually pre-defined modules. It uses the layout to instantiate modular networks. However, the modules are pre-defined which means the reasoning procedure does not produce new modules or relations anymore. As a result, it is difficult to meet the requirements of diversity of relations in dynamic and multi-step reasoning.

This paper tries to construct a chain of reasoning (CoR) for addressing these problems. Both of the iteratively updated relations and compound objects are used as nodes in the chain. Updated relations push reasoning to involve more compound objects; compound objects maintain the intermediate conclusions of reasoning and make the next-step relational reasoning possible by lowering the computational complexity efficiently. An example of the CoR is shown in Fig. 1. Initial objects in the image are first recognized, such as two buses and a sign in the original image. All pairwise relations between these objects are then calculated, and a combination of the relations are used to generate compound objects, such as *"bus on the right."* More complex relations are further calculated between the compound objects and initial objects to generate more complex compound objects, such as *"sign next to the bus on the right,"* which brings us the answer.

In summary, our contributions are as follows:

- We introduce a new VQA model that performs a chain of reasoning, which generates new relations and compound objects dynamically to infer the answer.
- We achieve new state-of-the-art results on four publicly available datasets. We conduct a detailed ablation study to show that our proposed chain structure is superior to stack structure and parallel structure.
- We visualize the chain of reasoning, which shows the progress that the CoR generates new compound objects dynamically that lead to the answer of the question step by step.

## 2 Related Work

Reasoning plays a crucial role in VQA. Recent studies modeled the reasoning procedure from different perspectives. In this section, we briefly review three types of existing work that enable reasoning. We also highlight differences between previous models and ours.

**Relation-based methods**    The relation-based method performs one-step relational reasoning to infer the answer. [4] proposed a plug-and-play module called "Relation Networks" (RN). RN uses full arrangement to model all the interactions between objects in the image and performs multi-layer perceptrons (MLPs) to calculate all the relations. Then, the relations are summed and passed through other MLPs to infer the final answer. Modeling pairwise relationships already brings the $O(m^2)$ computational complexity and makes it impossible to carry out multi-step reasoning. By object refining, our model lowers the computational complexity and makes the multi-step reasoning possible.

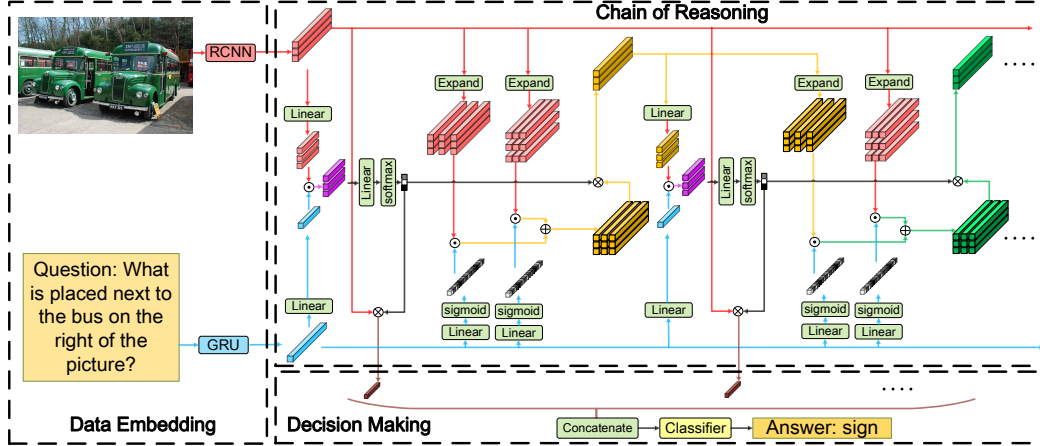

Figure 2: The overall structure of the proposed model for solving the VQA task. It consists of Data Embedding, Chain of Reasoning, and Decision Making, marked with dash lines respectively.

**Attention-based methods** Usually, attention-based methods enable reasoning by locating relevant objects in original input features, such as bounding boxes or image regions. Initially, [10] proposed one-step attention to locate relevant objects of images. Furthermore, [5, 6] proposed multi-step attention to update relevant objects of images and infer the answer progressively. Additionally, [11, 12] proposed multi-modal attention, which finds not only the relevant objects of images but also questions or answers. Recently, [13, 14, 15, 16] used bilinear fusion in attention mechanism to find more accurate objects of input features. Attention distributions in the above work are always on original input features. In contrast, our model pay attentions on not only objects in original input features but also new compound objects generated dynamically during reasoning.

**Module-based methods** Module-based methods try to define relations as modules in advance, and the reasoning procedure is determined by a layout generated from these modules. [7] proposed neural module network, which uses fixed layouts generated from dependency parses. Later, [8] proposed dynamic neural module network, which learns to optimize the layout structure by predicting a list of layout candidates. However, the layout candidates are still generated by dependency parses. To solve this problem, [9] proposed an end-to-end module network, which learns to optimize over full space of network and requires no parser at evaluation time. Our model forms new relations dynamically in the reasoning procedure, instead of choosing from a set of manually pre-defined modules.

## 3    Chain of Reasoning based model for VQA

The overall structure of our model for VQA is illustrated in Fig. 2. It consists of three parts: Data Embedding, Chain of Reasoning, and Decision Making. Data Embedding pre-processes the image and question. Chain of Reasoning is the core part of the model. Starting from outputs of Data Embedding, relational reasoning on initial objects forms new relations, and object refining generates new compound objects based on the new relations. Iteratively, these two operations on updated relations and objects build the chain of reasoning, which outputs a series of results. Decision Making makes use of all the results to select the final answer of the question. We give the details of the three parts in Section 3.1∼3.3 respectively.

### 3.1    Data Embedding

Faster-RCNN [17] is used to encode images with the static features provided by bottom-up-attention [18], GRU [19] is used to encode text with the parameters initialized with skip-thoughts [20], as denoted in Eq. (1).

$$V = RCNN(image), \quad Q = GRU(question), \tag{1}$$

where $V \in \mathbb{R}^{m \times d_v}$ denotes the visual features of the top-ranked $m$ detection boxes and $Q \in \mathbb{R}^{d_q}$ denotes the question embedding. Here, $V$ is viewed as a set of $m$ initial objects, i.e. $V = \{v_1, v_2, \ldots, v_m\}$. From the perspective of reasoning, $V$ can also be viewed as $m$ initial premises.

## 3.2 Chain of Reasoning

Starting from initial objects $O^{(1)} = V$ defined in Eq. (1), a chain of reasoning consists of a series of sub-chains and an output at each time, which is explained in Fig. 3.

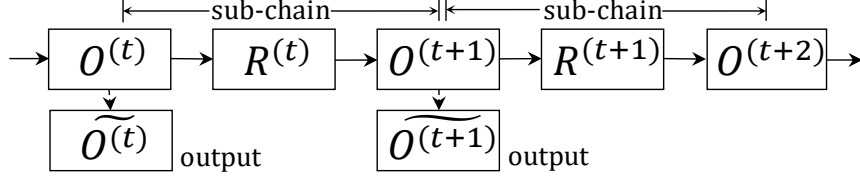

Figure 3: Sub-chains and their outputs in Chain of Reasoning.

In Fig. 3, $O^{(t)} \in \mathbb{R}^{m \times d_v}$ is the set of initial objects at time $t = 1$ or compound objects at time $t > 1$. $\widetilde{O^{(t)}} \in \mathbb{R}^{d_v}$ is the output of the chain at time $t$. $R^{(t)} \in \mathbb{R}^{m \times m \times d_v}$ is the set of updated relations at time $t$. $O^{(t+1)} \in \mathbb{R}^{m \times d_v}$ is the set of new compound objects at time $t + 1$. From the perspective of reasoning, $O^{(t)}$ can also be viewed as intermediate conclusions when $t > 1$. We first give the details on the output at time $t$, and then describe how the sub-chain is formed.

The output at time $t$ is designed to capture information provided by $O^{(t)}$ under the guidance of question . An attention-based method is used as in Eq. (2)~(5).

$$P^{(t)} = relu(O^{(t)} W_o^{(t)}), \quad S^{(t)} = relu(Q W_q^{(t)}), \tag{2}$$

$$F^{(t)} = \sum_{k=1}^{K} (P^{(t)} W_{p,k}^{(t)}) \odot (S^{(t)} W_{s,k}^{(t)}) \tag{3}$$

$$\alpha^{(t)} = softmax(F^{(t)} W_f^{(t)}), \tag{4}$$

$$\widetilde{O^{(t)}} = \left( \alpha^{(t)} \right)^{\mathsf{T}} O^{(t)}, \tag{5}$$

where Eq. (2) maps the objects at time $t$ to $P^{(t)} \in \mathbb{R}^{m \times d_p}$ and maps the question feature to $S^{(t)} \in \mathbb{R}^{d_s}$ at time $t$. Eq. (3) uses the Mutan fusion mechanism proposed by [16]. $K$ is the hyperparameter. $F^{(t)} \in \mathbb{R}^{m \times d_f}$ is the fusion embedding at time $t$. $\alpha^{(t)} \in \mathbb{R}^m$ in Eq. (4) is the attention distribution over the $m$ compound objects at time $t$. $\widetilde{O^{(t)}} \in \mathbb{R}^{d_v}$ in Eq. (5) is the result of attention at time $t$, which is also the output of chain of reasoning at time $t$. The output at each time $t$ will be used for final decision making. To write simple, we omit the bias $b$.

The sub-chain $O^{(t)} \to R^{(t)} \to O^{(t+1)}$ is performed in two operations. The first operation from $O^{(t)}$ to $R^{(t)}$ is called relational reasoning which forms new relations between objects, and the second operation from $R^{(t)}$ to $O^{(t+1)}$ is called object refining which generates new compound objects to start a new sub-chain. We introduce them respectively as follows.

**Relational reasoning from $O^{(t)}$ to $R^{(t)}$** The $m$ objects in $O^{(t)}$ interact with the $m$ initial objects in $O^{(1)}$ under the guidance of the question $Q$, as denoted in Eq. (6)~(7).

$$G_l = \sigma \left( relu(Q W_{l_1}) W_{l_2} \right), \quad G_r = \sigma \left( relu(Q W_{r_1}) W_{r_2} \right), \tag{6}$$

$$R_{ij}^{(t)} = (O_i^{(t)} \odot G_l) \oplus (O_j^{(1)} \odot G_r), \tag{7}$$

where Eq (6) maps question feature to the same dimension as the object feature by a two-layer MLP with different weights respectively. $\sigma$ is the sigmoid function. $G_l, G_r \in \mathbb{R}^{d_v}$ are the guidances. Eq. 7 is the sum of the guided $ith$ compound object at time $t$ and the guided $jth$ initial object. $\odot$ denotes the element-wise multiplication and $\oplus$ denotes the element-wise summation.

Notice that $G_l$ and $G_r$ are different guidances with different weights, but the weights in $G_l$ and $G_r$ are shared among all sub-chains respectively. As a result, two sets of weights are trained: the set of weights in $G_l$ make the question focus on the compound objects and another set of weights in $G_r$ make the question focus on the initial objects. This is in line with the reasoning procedure — the question decides what the model should do for the intermediate conclusions it already got and the initial premises. Besides, initial objects $O_j^{(1)}$ used at each time allow the model to capture initial premises through the whole reasoning procedure.

**Object refining from $R^{(t)}$ to $O^{(t+1)}$**    The previous relational reasoning operation produces $m \times m$ relations between $m$ compound objects and $m$ initial objects. Since modeling the pairwise relations increases the compexity of reasoning from $O(m)$ to $O(m^2)$, $n$-step reasoning will face the complexity of $O(m^n)$. In order to avoid the exponential complexity of multi-step reasoning, we refine these relations to $m$ new compound objects, each denoted in Eq. (8):

$$O_j^{(t+1)} = \sum_{i=1}^{m} \alpha_i^{(t)} R_{ij}^{(t)}, \tag{8}$$

where $O_j^{(t+1)}$ is the $j$th compound object at time $t+1$, In Eq. (8), the attention weights of the compound objects $\alpha^{(t)}$ are used to refine the relations $R^{(t)}$ formed by the compound objects and the initial objects. This has two advantages: Firstly, Eq. (8) is more in line with the reasoning procedure. The $j$th compound object at time $t+1$ is determined by all the compound objects at time $t$ and the $j$th initial object. This means that any conclusion generated by the next reasoning step will use all the intermediate conclusions in the previous step. At the same time, if an intermediate conclusion in the previous step is important, then its information is more likely to be used in the next step. Secondly, Eq. (8) makes it mathematically simple and computationally feasible to begin a next turn reasoning. Mathematically, we can use a single set of equations to describe the whole chain. Computationally, we can keep the complexity of $O(nm^2)$ when we perform $n$ sub-chains of reasoning.

## 3.3   Decision Making

The decision maker at time $T$ gives an answer to the question by making use of all the outputs $\widetilde{O^{(t)}}$ ($t = 1, 2, ..., T$). An concatenation is employed for integrating $T$ outputs in Eq. (9).

$$O^* = [relu(\widetilde{O^{(1)}}W^{(1)}); relu(\widetilde{O^{(2)}}W^{(2)}); ...; relu(\widetilde{O^{(T)}}W^{(T)})], \tag{9}$$

where $O^* \in \mathbb{R}^{d_*}$ is the joint feature of outputs. We further fuse joint feature and question by Eq.(10).

$$H = \sum_{k=1}^{K} (O^* W_{o^*,k}) \odot (Q W_{q',k}), \tag{10}$$

where $K \in \mathbb{R}^+$ is the hyperparameter, $H \in \mathbb{R}^{d_h}$ is the joint embedding. Finally, a linear layer with a softmax activation function is used to predict the candidate answer distribution as shown in Eq. (11).

$$\hat{a} = softmax(H W_h), \tag{11}$$

## 3.4   Training

We first calculate the ground-truth answer distribution in Eq. (12):

$$a_i = \frac{\sum_{j=1}^{N} \mathbb{1}\{u_j = i\}}{N - \sum_{j=1}^{N} \mathbb{1}\{u_j \notin \mathcal{D}\}}, \tag{12}$$

where $a \in \mathbb{R}^{|\mathcal{D}|}$ is the ground-truth answer distribution, $u_i$ is the answer given by the $i$th annotator. $N$ is the number of annotators. In detail, $N$ is 10 in the VQA 1.0 and VQA 2.0 dataset; $N$ is 1 in the COCO-QA dataset and the TDIUC dataset.

Finally, we use the KL-divergence as the loss function between $a$ and $\hat{a}$ in Eq. (13):

$$\mathcal{L}(\hat{a}, a) = \sum_{i=1}^{|\mathcal{D}|} a_i \log \left( \frac{a_i}{\hat{a}_i} \right). \tag{13}$$

Table 1: Comparision with the state-of-the-arts on the VQA 1.0 dataset.

| | Method | VQA 1.0 Test-dev | | | | | VQA 1.0 Test-std | | | | |
|---|---|---|---|---|---|---|---|---|---|---|---|
| | | Open-Ended | | | | MC | Open-Ended | | | | MC |
| | | All | Y/N | Num. | Other | All | All | Y/N | Num. | Other | All |
| Single image feature | HighOrderAtt[12] | - | - | - | - | 69.4 | - | - | - | - | 69.3 |
| | MLB(7)[14] | 66.77 | 84.54 | 39.21 | 57.81 | - | 66.89 | 84.61 | 39.07 | 57.79 | - |
| | Mutan(5)[16] | 67.42 | 85.14 | 39.81 | 58.52 | - | 67.36 | 84.91 | 39.79 | 58.35 | - |
| Multi image feature | DualMFA[21] | 66.01 | 83.59 | 40.18 | 56.84 | 70.04 | 66.09 | 83.37 | 40.39 | 56.89 | 69.97 |
| | ReasonNet[22] | - | - | - | - | - | 67.9 | 84.0 | 38.7 | **60.4** | - |
| Single image feauture | CoR-2(36boxes) (ours) | 68.16 | 85.57 | 43.76 | 58.80 | 72.60 | 68.19 | 85.61 | 43.10 | 58.75 | 72.61 |
| | CoR-3(36boxes) (ours) | **68.37** | **85.69** | **44.06** | **59.08** | **72.84** | **68.54** | **85.83** | 43.93 | 59.11 | **72.93** |

Table 2: Comparision with the state-of-the-arts on the VQA 2.0 dataset.

| Method | VQA 2.0 Test-dev | | | | VQA 2.0 Test-std | | | |
|---|---|---|---|---|---|---|---|---|
| | All | Y/N | Num. | Other | All | Y/N | Num. | Other |
| MF-SIG-VG[23] | 64.73 | 81.29 | 42.99 | 55.55 | - | - | - | - |
| Up-Down(36 boxes)[24] | 65.32 | 81.82 | 44.21 | 56.05 | 65.67 | 82.20 | 43.90 | 56.26 |
| LC_Baseline(100 boxes)[25] | 67.50 | 82.98 | 46.88 | 58.99 | 67.78 | 83.21 | 46.60 | 59.20 |
| LC_Counting(100 boxes)[25] | 68.09 | 83.14 | **51.62** | 58.97 | 68.41 | 83.56 | **51.39** | 59.11 |
| CoR-2(36 boxes) (ours) | 67.96 | 84.7 | 47.1 | 58.42 | 68.15 | 84.82 | 46.8 | 58.52 |
| CoR-3(36 boxes) (ours) | 68.19 | 84.98 | 47.19 | 58.64 | 68.59 | 85.16 | 47.19 | 59.07 |
| CoR-3(100 boxes) (ours) | **68.62** | **85.22** | 47.95 | **59.15** | **69.14** | **85.76** | 48.4 | **59.43** |

Table 3: Comparision with the state-of-the-arts on the COCO-QA dataset.

| Method | All | Obj. | Num. | Color | Loc. | WUPS0.9 | WUPS0.0 |
|---|---|---|---|---|---|---|---|
| QRU [26] | 62.50 | 65.06 | 46.90 | 60.50 | 56.99 | 72.58 | 91.62 |
| HieCoAtt [11] | 65.4 | 68.0 | 51.0 | 62.9 | 58.8 | 75.1 | 92.0 |
| Dual-MFA [21] | 66.49 | 68.86 | 51.32 | 65.89 | 58.92 | 76.15 | 92.29 |
| CoR-2(36 boxes) (ours) | 68.67 | 69.76 | 55.14 | 73.36 | 59.52 | 77.47 | 92.68 |
| CoR-3(36 boxes) (ours) | **69.38** | **70.42** | **55.83** | **74.13** | **60.57** | **78.10** | **92.86** |

Table 4: Comparision with the state-of-the-arts on the TDIUC dataset.

| Question Type | MCB-A[13] | RAU[27] | CATL-QTA$^W$[28] | CoR-2 (ours) | CoR-3 (ours) |
|---|---|---|---|---|---|
| Sceen Recognition | 93.06 | 93.96 | 93.80 | 94.48 | **94.68** |
| Sport Recognition | 92.77 | 93.47 | 95.55 | **95.94** | 95.90 |
| Color Attributes | 68.54 | 66.86 | 60.16 | 73.59 | **74.47** |
| Other Attributes | 56.72 | 56.49 | 54.36 | 59.59 | **60.02** |
| Activity Recognition | 52.35 | 51.60 | 60.10 | 60.29 | **62.19** |
| Positional Reasoning | 35.40 | 35.26 | 34.71 | 39.34 | **40.92** |
| Sub. Object Recognition | 85.54 | 86.11 | 86.98 | 88.38 | **88.83** |
| Absurd | 84.82 | 96.08 | **100.00** | 95.17 | 94.70 |
| Utility and Affordances | 35.09 | 31.58 | 31.48 | **40.35** | 37.43 |
| Object Presence | 93.64 | 94.38 | 94.55 | 95.40 | **95.75** |
| Counting | 51.01 | 48.43 | 53.25 | 57.72 | **58.83** |
| Sentiment Understanding | 66.25 | 60.09 | 64.38 | 66.72 | **67.19** |
| Overall (Arithmetric MPT) | 67.90 | 67.81 | 69.11 | 72.25 | **72.58** |
| Overall (Harmonic MPT) | 60.47 | 59.00 | 60.08 | 65.65 | **65.77** |
| Overall Accuracy | 81.86 | 84.26 | 85.03 | 86.58 | **86.91** |

# 4 Experiments

## 4.1 Datasets and evaluation metrics

We evaluate our model on four public datasets: the VQA 1.0 dataset [29], the VQA 2.0 dataset [30], the COCO-QA dataset[31] and the TDIUC dataset [27]. VQA 1.0 contains 614,163 samples, including 204,721 images from COCO [32]. VQA 2.0 is a more balanced version and contains 1,105,904 samples. COCO-QA is a smaller dataset that contains 78,736 samples. TDIUC is a larger dataset that contains 1,654,167 samples and 12 question types. For VQA 1.0 and VQA 2.0, we use the evaluation tool proposed in [29] to evaluate the model. For COCO-QA and TDIUC, we calculate the simple accuracy for each question type. Besides, additional WUPS [33] is calculated for COCO-QA and additional Arithmetic/Harmonic mean-per-type (MPT) [27] is calculated for TDIUC.

## 4.2 Implementation details

During the data-embedding phase, the image features are mapped to the size of $36 \times 2048$ and the text features are mapped to the size of 2400. In the chain of reasoning phase, the number of hidden layer in Mutan is 510; hyperparameter $K$ is 5. The attention hidden unit number is 620. In the decision making phase, the joint feature embedding is set to 510. All the nonlinear layers of the model all use the relu activation function and dropout [34] to prevent overfitting. All settings are commonly used in previous work. We implement the model using Pytorch. We use Adam[35] to train the model with a learning rate of $10^{-4}$ and a batch_size of 64. More details, including source codes, will be published in the near future.

## 4.3 Comparison with the state-of-the-art

In this section, we compare our single CoR-$T$ model with the state-of-the-art models on four datasets. CoR-$T$ means that the model consists of $T$ sub-chains. Firstly, Tab. 1 shows the results on the VQA 1.0 dataset. Using a single image feature, CoR-3 not only outperforms all the models that use single image feature but also outperforms the state-of-the-art ReasonNet [22] model, which uses six input image features including face analysis, object classification, scene classification and so on. Secondly, Tab. 2 shows the results on the VQA 2.0 dataset. Compared with Up-Down (36 boxes) [24], which is the winning model in the VQA challenge 2017, CoR-3 (36 boxes) achieves 2.92% higher accuracy in test-std set. Compared with the most recent state-of-the-art model LC_counting (100 boxes) [25], our single CoR-3 (100 boxes) model achieves a new state-of-the-art result of 69.14% in the test-std set. Thirdly, Tab. 3 shows the results on the COCO-QA dataset. CoR-3 improves the overall accuracy of the state-of-the-art Dual-MFA from 66.49% to 69.38%. In particular, there is an improvement of 4.51% in *"Num."* and 8.24% in *"Color"*. Fourthly, Tab. 4 shows the results on the TDIUC dataset. CoR-3 improves the overall accuracy of the state-of-the-art CATL-QTA$^W$ [28] from 85.03% to 86.91%. There is also an improvement of 5.58% in *"Counting"* and 5.93% in *"Color Attributes"*. In summary, CoR achieves consistently best performance on all four datasets.

## 4.4 Ablation study

In this section, we conduct some ablation experiments. For a fair comparion, all the data provided in this section are trained under the VQA 2.0 training set and tested on the VQA 2.0 validation set. All the models use the exact same bottom-up-attention feature (36 boxes) extracted from faster-rcnn.

Tab. 5 shows the effectiveness of the chain structure. We implement MLB[14], Mutan [16] and their stack and parallel structure. The stack structure is proposed by SAN [5], which stacks 2 or 3 attention layers. The parallel structure is similar to Multi-Head Attention [36], which consists of 2 or 3 attention layers running in parallel. As shown in Tab. 5, the chain structure not only significantly improves the performance of attention models but also superior to their stack or parallel structures. For example, compared with Mutan, Mutan-Stack-3 is only 0.29% higher while CoR-3 is 1.53% higher. Furthermore, the chain structure is insensitive to the attention model. CoR-2 and CoR-3 can achieve high performance whether using Mutan or MLB.

Tab. 6 shows the effectiveness of the relational reasoning operation. Firstly, we implement CoR-2 with $[O_i^{(t)}; O_j^{(1)}; G]W_1$, which is proposed RN [4]. We find it lowers the performance (64.96%→62.46%). This is because the purpose of relational reasoning here is to prepare for generating compound

Table 5: Effectiveness of the chain structure on the VQA 2.0 validation.

| Method | MLB[14] | MLB-Stack-2 | MLB-Stack-3 | MLB-Parallel-2 | MLB-Parallel-3 | CoR-2 with MLB | CoR-3 with MLB |
|---|---|---|---|---|---|---|---|
| Val | 62.91 | 63.28 | 63.55 | 63.20 | 63.28 | 64.90 | **64.96** |

| Method | Mutan[16] | Mutan-Stack-2 | Mutan-Stack-3 | Mutan-Parallel-2 | Mutan-Parallel-3 | CoR-2 | CoR-3 |
|---|---|---|---|---|---|---|---|
| Val | 63.61 | 63.78 | 63.90 | 63.66 | 63.80 | 64.96 | **65.14** |

Table 6: Effectiveness of relational reasoning operation on the VQA 2.0 validation.

| Method | Val |
|---|---|
| CoR-2 with $[O_i^{(t)}; O_j^{(1)}; G]W_1$ | 62.46 |
| CoR-2 with $(O_i^{(t)} + O_j^{(1)}) \odot G$ | 64.73 |
| CoR-2 with $(O_i^{(t)} \odot G_l) \oplus (O_j^{(t)} \odot G_r)$ | 64.24 |
| CoR-2 | **64.96** |

Table 7: Effectiveness of object refining operation on the VQA 2.0 validation.

| Method | Val |
|---|---|
| CoR-2 with $\sum_{i=1}^{m} \alpha_i^{(t)} R_{ji}^{(t)}$ | 64.42 |
| CoR-2 | **64.96** |

Table 8: Effectiveness of the model on different question types on the CLEVR dataset.

| Method | Overall | Count | Exist | Compare Numbers | Query Attribute | Compare Attribute |
|---|---|---|---|---|---|---|
| MLB | 85.0 | 90.0 | 76.7 | 78.8 | 91.1 | 82.7 |
| Mutan | 86.3 | 92.5 | 80.2 | 81.7 | 91.2 | 84.5 |
| RN | 96.4 | - | - | - | - | - |
| CoR-2 | **98.7** | **98.8** | **97.7** | **92.3** | **99.9** | **99.7** |

objects, and the element-wise sum in Eq. (7) is more fine-grained. Secondly, we implement CoR-2 with $(O_i^{(t)} \oplus O_j^{(1)}) \odot G$, which uses a single question guidance and also lowers performance (64.96%→64.73%). This shows that different guidances for compound objects and initial objects are beneficial to improve the performance. Thirdly, we implement CoR-2 with $(O_i^{(t)} \odot G_l) \oplus (O_j^{(t)} \odot G_r)$, which calculates the relations by the compound objects themselves without the initial object $O_j^{(1)}$. We find it still lowers the performance (64.96%→64.24%). This shows using initial premises $O_j^{(1)}$ at each step is crucial and may avoid "over-reasoning" by modeling very complex relations between compound objects.

Tab. 7 shows the effectiveness of the object refining operation. We implement a similar operation $\sum_{i=1}^{m} \alpha_i^{(t)} R_{ji}^{(t)}$. Although the formula is similar to $\sum_{i=1}^{m} \alpha_i^{(t)} R_{ij}^{(t)}$ in Eq. 8, the meaning is totally different. $\sum_{i=1}^{m} \alpha_i^{(t)} R_{ij}^{(t)}$ generates the $j$th compound object by weighted sum of the relations between each compound object and the $j$th initial object while $\sum_{i=1}^{m} \alpha_i^{(t)} R_{ji}^{(t)}$ generates that by weighted sum of the relations between each initial object and $j$th compound object. The former focuses on using the previous reasoning conclusions while the latter focuses on the initial premises. CoR-2 has better results and is more in line with the reasoning procedure — focusing more on previous intermediate conclusions to push the next step reasoning.

Tab. 8 shows effectiveness of the model on different question types. We conduct experiments on the state description matrix version of the CLEVR dataset [37]. CoR-2 reaches an overall accuracy of 98.7%, which outperforms MLB and Mutan on the same setup. Furthermore, CoR-2 achieves the performance of 99.9% in question type of *"Query Attribute"* and 99.7% in question type of *"Compare Attribute"*. It is worth mentioning that there is still room for improvement in *"Compare Numbers"* questions.

## 4.5 Qualitative evaluation

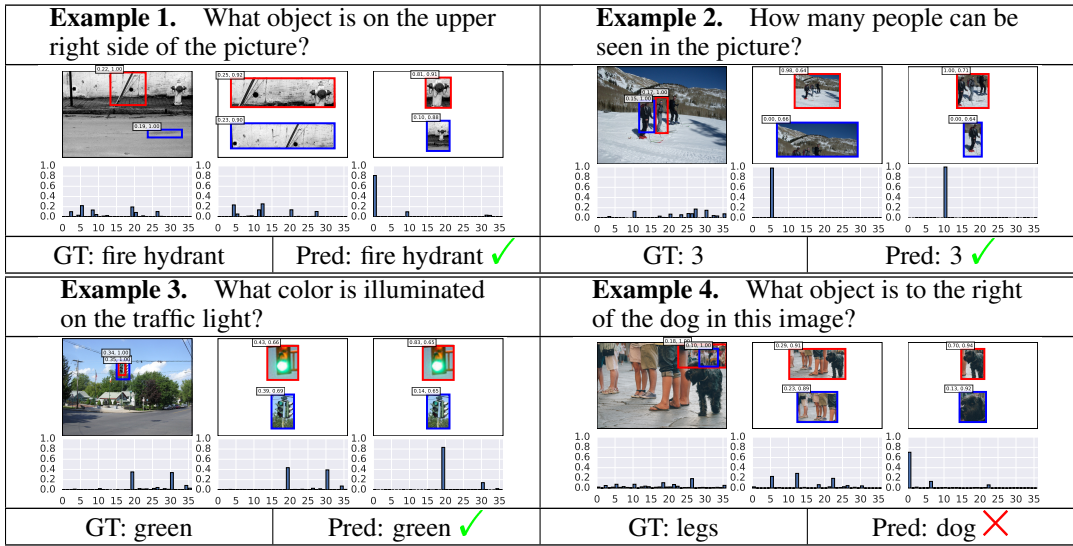

Figure 4: Visualization of the reasoning procedure of CoR-3.

In Figure 4, we visualize the compound objects generated by CoR-3 and their attention weights. Four examples are given including three success cases and one failure case. Each example contains three steps. The red box and the blue box in each step represent objects with the top two attention weights respectively. The initial objects in the first step are part of the original image and easy to visualize by the bounding box, but the compound objects in the second and third step are difficult to visualize directly. Therefore, we search from $1105904 \times 36$ boxes (1105904 is the number of samples and each sample has 36 boxes) and find the box with the most similar feature by cosine similarity to represent the compound object. The upper left corner of each box contains a tuple of the form $(w, s)$. $w$ is the attention weight, $s$ is the similarity between the searched box and the real compound object.

In **Example 1**, the left image shows a pillar (red box) and ground (blue box). Their values of $w$ are 0.22 and 0.19 respectively. Since they are initial top two rcnn objects in $O^{(1)}$, the values of $s$ are 1. The model focuses on some disperse *"objects"*, which can be further seen by attention distribution histogram below. The middle image shows top two compound objects in the second step. The red box focuses on *"objects on the upper"*. The attention weight of the red box increased slightly to 0.25. The similarity between the red box and the real compound object is 0.92. The right image shows top two more complex compound objects in the third step. The *"objects on the upper right"* has been focused in the red box. Interestingly, the $w$ of the red box increases to 0.81, which means in the third step, CoR-3 is very confident that the box containing *"hydrant"* is exactly the final answer. Statistics show that 96.76% of the success cases satisfy the phenomenon of dispersion to concentration. In **Example 2~3**, two more success cases are shown. In **Example 4**, the model already gets the intermediate result *"dog in the image"* in the third step but fails to further find *"leg on the right of the dog in the image"*, which seems that three-step reasoning is insufficient here.

## 5 Conclusion

In this paper, we propose a novel chain of reasoning model for VQA task. The reasoning procedure is viewed as the alternate updating of objects and relations. Experimental results on four publicly available datasets show that CoR outperforms state-of-the-art approaches. Ablation study shows that proposed chain structure is superior to stack structure and parallel strucure. The visualization of the chain of reasoning illustrates the progress that the CoR generates new compound objects that lead to the answer of the question step-by-step. In the future, we plan to apply CoR to other tasks that require reasoning like reading comprehension question answering or video question answering.

**Acknowledgments**

We would like to thank the anonymous reviewers for their valuable comments. This paper is supported by NSFC (No. 61273365), NSSFC (2016ZDA055), 111 Project (No. B08004), Beijing Advanced Innovation Center for Imaging Technology, Engineering Research Center of Information Networks of MOE, China.

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
