[Reviews · NeurIPS 2018]

Reviewer 1



Paper Summary: This paper presented a novel approach that performs chain of reasonings on the object level to generate answer for visual question answering. Object-level visual embeddings are first extracted through object detection networks as visual representation and sentence embedding of the question are extract question representation. Based on these, a sequential model that performs multi-steps of relational inference over (compound) object embeddings with the guidance of question is used to obtain the final representation for each sub-chain inference. A concatenation of these embeddings are then used to perform answer classification. Extensive experiments have been conducted on four public datasets and it achieves state-of-the-art performance on all of them. The paper is well written and easy to follow in most parts. Extensive experiments has been done to thoroughly studied each component of the proposed algorithm. Qualitative visualization in the section 4.5 provides an intuitive understanding about the model's behavior of reasoning. Minor Comments: 1. In the equation (10), it seems there is no subscript k in the main body of the equation? What does the k in summation refer to? 2. It seems like some implementation details are missed in both the main paper and supplementary material. For example, does the Faster RCNN need pre-training? On which dataset? Is the final answer classification network predicting the top-frequent answers or entire answer space? Although authors guarantee to make source code available, it would be nice to list those specific details in somewhere (e.g supplementary material ). 3. It would be nice to see some results on CLEVR, to evaluate model's performance based on different types of the questions, and also its performance on the CLEVR's compositional split, as a diagnosis of model's capability. Post Rebuttal Feedback: Thanks for the feedback. I believe the authors have addressed most of my concerns raised in the review and glad to see that CoR2 outperforms relational networks on CLEVR. It would be great if authors could report detailed results with respect to the question types on the CLEVR. I believe that a revised version that fix most concerns reviewers has raised (e.g. CELVR results, implementation details, self-contained content, typos, etc) would deserve an acceptance decision.

Reviewer 2



Summary: The authors propose a new VQA model that combines gated object features provided by bottom-up attention. The object features are gated using the question feature and combined multiple times forming compound object features. It achieves the state of the art performance on multiple VQA benchmarks. Strength: 1. The biggest strength is the outstanding performance. It consistently outperforms previous approaches. 2. The idea of the interaction between objects is interesting. Applying such technique multiple times is also well-motivated. Weakness and questions: 1. It seems like the model is not capable of handling the interaction between compound objects since the model only merges the compound objects with the initial objects in Eq. (7). Does the model somehow handle such cases? 2. I am not convinced to call the procedure of computing R^{(t)} as relational reasoning because the model only elementwise-sums the gated object features. The resulting feature R^{(t)} can only contain the element-wise linear combination of the initial object features for any time t. 3. Reusing the gating networks for multiple time steps is not intuitive. Since both gating networks for computing R^{(t)} using the weights W_{l_1}, W_{l_2}, W_{r_1}, W_{r_2} takes the question as its only input, the same gating weights G_l and G_r would be obtained at every time step. In authors' rebuttal, I would like to see the answers to the above points in weakness. Especially, I believe that clear answers to the second and third points are necessary for the technical novelty of the paper. After the rebuttal phase: I thank the authors for the rebuttal answering all my questions. The rebuttal mainly presents empirical results to show that the proposed method performs better than the methods in the questions. However, I am still not fully convinced by their arguments as there are many possible ways to incorporate such concepts (eg., we can adopt a recurrent architecture to avoid the linearly increasing number of parameters while maintaining different gating values at different time steps.). I believe that these can be good directions to improve the method in their future work. Despite some of my concerns, I still think it is worth to accept this paper considering the contributions (written in the strength section of this review) with the reasonable empirical supports.

Reviewer 3



*Summary* Questions in Visual Question Answering often require to reason about referring expressions that relate some object to other objects in the image, e.g. “The bus on the right of the person that is wearing a hat”. This paper argues that previous architectures for VQA are not explicitly biased to capture such chains of reasoning and propose a novel architecture that interleaves a relational reasoning step (to form novel relationships between objects) with a refinement operator (to prevent exponential blow-up). The model improves the state-of-the-art on four VQA datasets: VQA v1 and v2, Coco QA and TDIUC. In the ablation studies, the authors show that the chain of reasoning structure outperforms the stack-based and parallel-based attention structure. *Originality* The observation that referring expressions are compositional and that neural networks for VQA should incorporate compositional priors is not novel (see e.g. compositional attention network [Hudson and Manning, ICLR2018]). Nevertheless, it is interesting that this paper applies a relational prior on top of extracted *object* features, as opposed to prior work that has proposed to apply it on top of convnet features [Relation Networks, Santoro et al]. Also, the relational reasoning module is different from prior work, and these details seem to matter; The ablation study highlights the benefit of the proposed module over a vanilla relation network module. *Clarity* The paper is well-structured but could benefit from an additional pass of proofreading. The model description is mathematically precise, and I believe one could reproduce the experimental results from the paper description. *Quality* Experiments are thoroughly executed, improving the state-of-the-art on four VQA datasets. Also, the ablation study compares the relative improvements of several hyperparameters of the proposed module, as well as relative improvements over related architectures. *Significance* The proposed model is strong performant, and could be widely applied to a range of language-vision tasks. Some of my remaining concerns: - Although evaluation is thorough, I believe some other datasets would be better suited to evaluate the proposed model. Instead of Coco QA, I would have suggested evaluating on CLEVR [Johnson et al, CVPR17] or the oracle task of GuessWhat [de Vries, CVPR17]. - In the ablation studies, the authors mention that the chain of reasoning structure outperforms the stacked attention baseline and parallel attention baseline. Although I remember the stacked attention network architecture by heart, I did not figure out what the parallel architecture would be, so maybe you can add a few lines to that paper to make it more self-contained? Post-rebuttal Feedback I'm keeping my scores.